# General In-Hand Object Rotation with Vision and Touch

**Haozhi Qi**[1,2]    **Brent Yi**[1]    **Sudharshan Suresh**[2,3]
**Mike Lambeta**[2]    **Yi Ma**[1]    **Roberto Calandra**[4,5]    **Jitendra Malik**[1,2]

[1]UC Berkeley    [2]Meta AI    [3]CMU    [4]TU Dresden
[5] The Centre for Tactile Internet with Human-in-the-Loop (CeTI)

https://haozhi.io/rotateit/

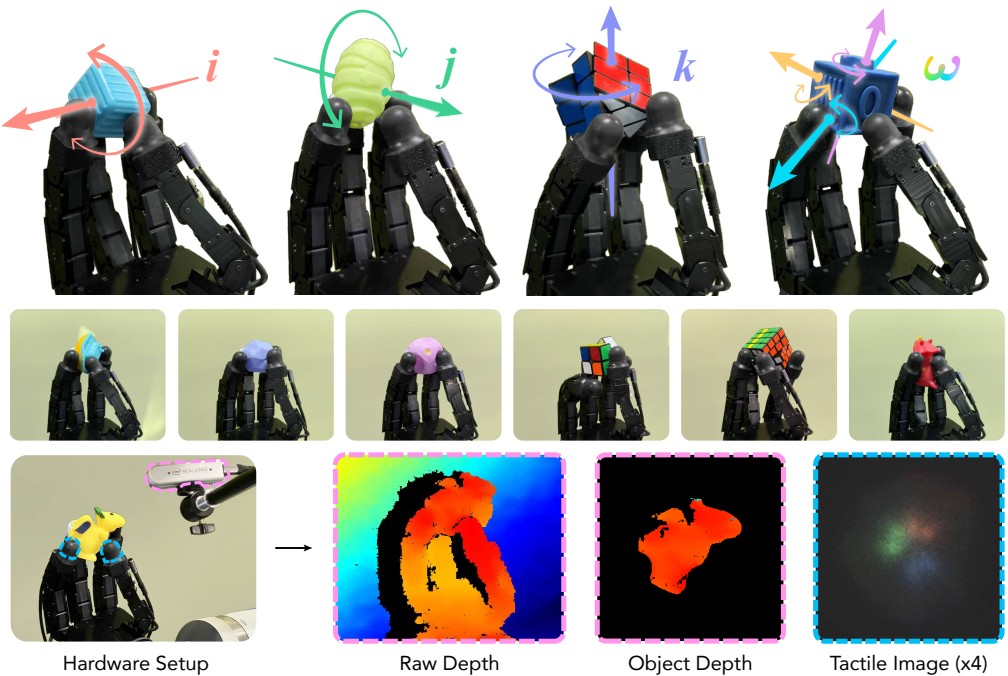

Figure 1: **Rotation over multiple axes by integrating proprioception, vision, and touch sensing.** *RotateIt* is trained in simulation and deployed directly to the real-world, where it generalizes to diverse test objects without the need for fine-tuning. Please see our website for more videos.

**Abstract:** We introduce *RotateIt*, a system that enables fingertip-based object rotation along multiple axes by leveraging multimodal sensory inputs. Our system is trained in simulation, where it has access to ground-truth object shapes and physical properties. Then we distill it to operate on realistic yet noisy simulated visuotactile and proprioceptive sensory inputs. These multimodal inputs are fused via a visuotactile transformer, enabling online inference of object shapes and physical properties during deployment. We show significant performance improvements over prior methods and the importance of visual and tactile sensing.

**Keywords:** In-Hand Object Rotation, Tactile Sensing, Reinforcement Learning, Sim-to-Real, Transformer, Visuotactile Manipulation

## 1    Introduction

Despite recent progress on in-hand manipulation for a single or a few objects [1, 2, 3, 4], generalizable object manipulation remains a challenge. In this paper, we present a model that integrates visual, tactile, and proprioceptive sensory inputs and achieves fingertip-based in-hand object rotation over multiple different axes. This continuous rotation task is important for achieving large-angle in-hand re-orientation skill and is challenging because it requires simultaneously maintaining stable force closure for objects with diverse geometries.

7th Conference on Robot Learning (CoRL 2023), Atlanta, USA.

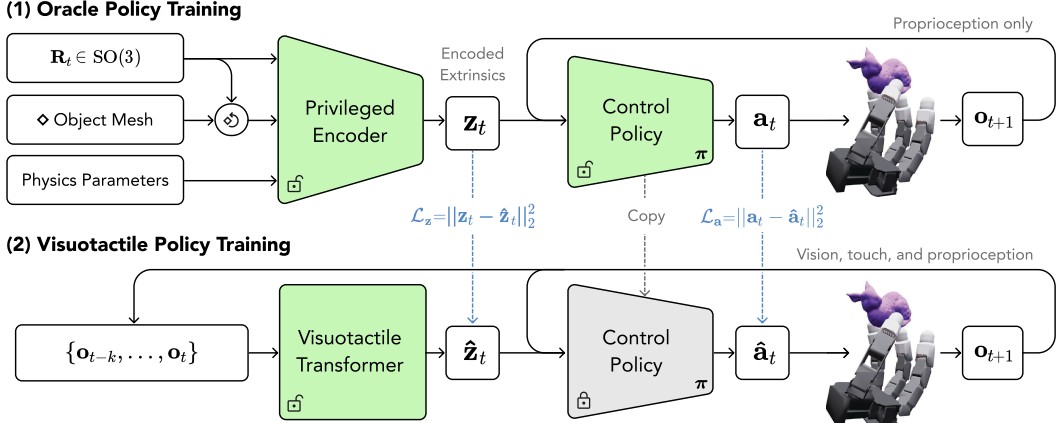

Figure 2: **An overview of our training pipeline.** Trainable components are highlighted in green. In oracle policy training, we jointly optimize the privileged encoder and control policy using PPO. In the visuotactile policy training, we feed a sequence of visuotactile and proprioceptive inputs to a transformer to infer $\hat{z}_t$. The visuotactile transformer is trained by minimizing the regression loss between $z_t$ and $\hat{z}_t$.

An overview of our method, *RotateIt*, is shown in Figure 2. Our approach draws inspiration from recent advances in training reinforcement learning policies with privileged information [5, 6, 7, 8], more specifically rapid motor adaptation [6, 7]. We first train an oracle policy that is conditioned on a representation of the privileged information (called extrinsics, denoted as $z_t$, as shown in Figure 2), which contains ground-truth physical properties and shapes of the objects. With access to this representation, the oracle policy is able to efficiently and stably manipulate diverse objects over multiple axes *in the simulator*.

The key challenge for real-world deployment lies in estimating the extrinsics encoding when privileged information is inaccessible. To address this challenge, we use multimodal sensing from vision and touch, just as humans do [9, 10, 11]. We implement this by designing a visuotactile transformer which operates on a history of multimodal proprioceptive, visual, and tactile inputs to infer $z_t$. Concretely, during training, we rollout the oracle policy in simulation and collect the foreground object depth, contact locations on the fingertips, proprioception, and action history. Then we feed these multimodal streams into a transformer to produce an estimate of the $z_t$, denoted as $\hat{z}_t$. The visuotactile transformer is trained to minimize the difference between the predicted and estimated encodings of the privileged information. In the real-world, we get the foreground objects using Segment Anything [12, 13], which enables *RotateIt* to be robust to cluttered backgrounds. We use tactile images from an omnidirectional vision-based touch sensor to retrieve these contact locations.

We demonstrate *RotateIt* can perform multi-axis object rotation using only its fingertips. In simulation, we quantitatively study the performance of rotating skills over three principal axes in the hand-centric frame and the impact of incorporating vision and touch at various stages (Section 5.1 and Section 5.2). To further understand what is learned by the policy, we investigate how accurately the latent representation of the policy captures objects' shapes by using it to recover 3D shapes (Section 5.3). Finally, we deploy the learned policies to rotate multiple different objects over multiple axes in the real world (Section 5.4). On the website, we show our policy can rotate objects, including but not limited to, the three canonical axes. Our work highlights the importance of both visual and tactile sensing in manipulation and presenting a step towards general dexterous in-hand manipulation.

## 2   Related Work

**Classic Control Methods**. Classical methods typically minimize a desired cost function using planning methods and a simplified system model [14, 15, 16, 17, 18, 19, 20]. State-of-the-art systems in this category include full $SO(3)$ reorientation using a compliance-enabled hand [21] and an accurate pose tracker [22]. In contrast, our work does not rely on an object model: we instead combine multi-sensory inputs with a learning-based policy that is trained on a large set of objects.

**Real-World Learning.** In-hand manipulation skills can be learned directly in the real-world, either by reinforcement learning [23, 24, 25] or imitation learning [26, 27, 28, 29]. However, reinforcement learning methods usually suffer from sample efficiency and real-world environment cannot provide enough variation. In contrast, our policy is trained using reinforcement learning via GPU-accelerated simulators, and does not need any human demonstrations.

**Sim-to-Real Methods.** OpenAI et al. [1, 2] first transferred dexterous in-hand manipulation policies to the real-world. Similarly, Sievers et al. [30], Pitz et al. [4] uses a torque-controlled hand for cube rotation and reorientation when the hand facing downwards. However, they focus on manipulating one single object. Although generalizable in-hand manipulation of diverse objects can be learned in simulation [31, 32, 33], transferring it to the real world remains a challenge.

Among sim-to-real methods, learning with privileged information [5, 6, 7, 34, 35] is shown to be effective for legged locomotion [5, 6] and manipulation [7, 8]. Recently, several works [7, 30, 36, 37] study generalizable in-hand object rotation using sim-to-real and reinforcement learning. In this paper, we use the same robot hand (Allegro [38]) as Qi et al. [7] and a similar variety of rotated objects with the significant advance being that the use of visual and tactile information now enables to rotate the object about an arbitrary axis, not just the $z$-axis. Other previous work [30, 37] shared this limitation of only demonstrating rotation about the $z$-axis. Compared to [36], our task is more challenging as it does not utilize a supporting surface, which allows constant tactile feedback on fingertips and enables a natural finger-gaiting to emerge. While we study continuous rotation (as many revolutions as possible), Chen et al. [8] study the task of object reorientation to an arbitrary pose (no limitation to $z$-axis rotation) and obtain impressive results. There are significant differences to our approach in oracle policy training, hand hardware, and goal specification. In addition to proprioception, they only use vision to sense the object, while our visuotactile policy utilizes both vision and touch. In the experiment section, we show that both these components improve the performance of our system.

**Visuotactile Sensing and Learning.** Tactile sensors such as GelSight [39], TacTip [40], DIGIT [41], DTact [42], GelTip [43], ArrayBot [44], and AllSight [45] have been used for numerous applications including grasping [46], playing the piano [47, 48], 3D reconstruction and localization [49, 50, 51, 52], and cup unstacking and bottle opening [53]. Previous work explores the usage of vision and touch for manipulation [54, 55, 56] but not for in-hand manipulation. To the best of our knowledge, *RotateIt* is the first work that intersects visuotactile sensing and learning to achieve general in-hand object rotation with a dexterous hand.

**Transformers in Robotics.** The transformer architecture [57] was originally proposed for machine translation and later used in computer vision [58]. In robotics, there are growing attempts to incorporate it in imitation learning [59, 60, 61, 62, 63] or reinforcement learning [64, 65, 66]. Chen et al. [67] also use the transformer on multimodal data but their tactile refers to force/torque sensing on robot joints. In contrast, our method uses the transformer for temporal modeling of multimodal proprioceptive, visual, and tactile information.

## 3 General In-hand Object Rotation with Vision and Touch

An overview of our method is shown in Figure 2. Our policy training consists of two stages: First, we train an *oracle policy* with privileged information. Next, we train a *visuotactile policy* with realistic yet noisy observations. Both of these stages happen *in simulation*. In this paper, we consider privileged information to be the object physical properties and object shape information. Real-world observations comprise of a stream of proprioceptive, visual, and tactile inputs. Our method trains one policy for each rotation axis and we show how to distill them into one general policy in Section A.2.

### 3.1 Oracle Policy Training

**Privileged Information.** For the object shape information, we sample $N_p$ points from the object's mesh and encode it to a feature vector $z_t^{\text{shape}}$ with $c_p$ dimensions using PointNet [68]. One key difference from previous works [7, 8] is that we explicitly encode object shape into the oracle policy, which we find to be critical especially for complex objects that are harder to manipulate.

| Method | x-axis | | | y-axis | | | z-axis | | |
|---|---|---|---|---|---|---|---|---|---|
| | RotR ↑ | TTF ↑ | RotP ↓ | RotR ↑ | TTF ↑ | RotP ↓ | RotR ↑ | TTF ↑ | RotP ↓ |
| Hora [7] | $79.13_{\pm 11.22}$ | $0.52_{\pm 0.02}$ | $0.55_{\pm 0.03}$ | $82.25_{\pm 14.21}$ | $0.54_{\pm 0.04}$ | $0.44_{\pm 0.01}$ | $99.83_{\pm 11.72}$ | $0.60_{\pm 0.03}$ | $0.39_{\pm 0.04}$ |
| **Oracle** | $\mathbf{125.23}_{\pm 16.24}$ | $\mathbf{0.79}_{\pm 0.03}$ | $\mathbf{0.35}_{\pm 0.02}$ | $\mathbf{118.26}_{\pm 13.20}$ | $\mathbf{0.79}_{\pm 0.05}$ | $\mathbf{0.30}_{\pm 0.01}$ | $\mathbf{140.90}_{\pm 17.26}$ | $\mathbf{0.82}_{\pm 0.02}$ | $\mathbf{0.27}_{\pm 0.01}$ |
| w/o shape | $85.10_{\pm 12.56}$ | $0.56_{\pm 0.03}$ | $0.39_{\pm 0.03}$ | $99.92_{\pm 10.21}$ | $0.62_{\pm 0.04}$ | $0.41_{\pm 0.02}$ | $129.38_{\pm 10.26}$ | $0.75_{\pm 0.03}$ | $0.29_{\pm 0.01}$ |

Table 1: We compare the performance improvement over various baselines on the rotation task over three different axes, under the same training setting. Compared to [7], we first add object and finger pose (w/o shape entry). This component slightly improves the performance. We then add object shape information into the oracle policy and this significantly improves the performance.

The physics property contains object's mass, center of mass, coefficient of friction, scale, and restitution, resulting in a 7-dimensional vector. The pose contains object's position, orientation (as a quaternion), and angular velocity, resulting a 10-dimensional vector. These vectors are concatenated together and projected to an 8-dim encoding vector $z_t^{\text{phys}}$. Our final privileged encoding is concatenated from the shape encoding and physical property encoding $z_t = [z_t^{\text{phys}}, z_t^{\text{shape}}]$.

**Observations and Outputs.** The oracle policy $\pi$ takes the robot's proprioception and the encoded privileged information $z_t$ as input. It outputs the targets of the PD Controller $a_t \in \mathbb{R}^{16}$. The observation $p_t$ contains a small temporal window of joint positions and actions $p_t = [q_{t-2:t}, a_{t-3:t-1}] \in \mathbb{R}^{96}$, where $q_t \in \mathbb{R}^{16}$ stands for the joint positions of the robot. Formally, we have $a_t = \pi(p_t, z_t)$.

**Reward Function.** Our reward function is modified from [7] with an additional penalty on undesired angular velocities component:

$$r \doteq r_{\text{rotr}} + \lambda_{\text{rotp}} r_{\text{rotp}} + \lambda_{\text{pose}} \, r_{\text{pose}} + \lambda_{\text{linvel}} \, r_{\text{linvel}} + \lambda_{\text{work}} \, r_{\text{work}} + \lambda_{\text{torque}} \, r_{\text{torque}} \, . \tag{1}$$

The object rotation task is defined as $r_{\text{rotr}} \doteq \max(\min(\boldsymbol{\omega} \cdot \boldsymbol{k}, r_{\max}), r_{\min})$ where $\boldsymbol{\omega}$ is the object's angular velocity and $\boldsymbol{k}$ is the desired rotation axis in the hand-centric axis. Naively applying this reward will result in unstable behaviors when rotating over $x$ and $y$-axis. To alleviate this problem, we add a rotation penalty term $r_{\text{rotp}} \doteq \|\boldsymbol{\omega} \times \boldsymbol{k}\|_1$. To make the policy stable, smooth, and energy efficient [6, 7, 69, 70], we use a few penalty terms: $r_{\text{pose}} \doteq - \|q - q_{\text{init}}\|_2^2$ is the hand pose deviation penalty, $r_{\text{torque}} \doteq - \|\boldsymbol{\tau}\|_2^2$ is the torque penalty, $r_{\text{work}} \doteq -\boldsymbol{\tau}^T \dot{\boldsymbol{q}}$ is the energy consumption penalty, and $r_{\text{linvel}} \doteq - \|\boldsymbol{v}\|_2^2$ is the object linear velocity penalty, where $q_{\text{init}}$ be the starting robot configuration, $\boldsymbol{\tau}$ be the commanded torques at each timestep, and $\boldsymbol{v}$ is the object's linear velocity.

**Policy Optimization.** We use PPO [71] to optimize the oracle policy. The weights between the policy and the critic network are shared, with an extra linear projection layer to estimate the value function. During training, each environment is assigned to an object with randomized physical properties and a stable initial grasp. We curate a list of hundreds of objects for training as shown in Figure 9.

### 3.2 Visuotactile Policy Training with Transformers.

We find robust and adaptive finger-gaiting emerges from the oracle policy training. However, it is assumed to know full object physical properties, pose, and shape as the input. To deploy it in the real-world, we need to use real-world observations to infer (representations of) these properties. Qi et al. [7] uses proprioceptive states to estimate such information. In this work, we augment it to include vision and touch and study their important roles in improving manipulation performance.

**Touch (Figure 3).** To reduce the sim-to-real gap for tactile sensors, we choose to use the discretized contact location projected on 2D plane as the proxy of tactile information. In simulation, we directly parse the contact position provided by the simulator, project it onto a 2D plane in fingertip frame, and discretize it to 8 locations. Specifically, the touch observation $o_t^{\text{touch}}$ is a $N_c \times 9$ dimensional array, where $N_c$ is the number of contact at each timestep. For each contact, it contains the discretized contact location (8-dimension) and the index of the finger. During training, since the number of contact points across timesteps are not the same, we use an MLP to each contact information and take an average of different contact point features. In the real-world, we use four omnidirectional vision-based touch sensors at the fingertips. We track the deformation of the highest intensity pixel

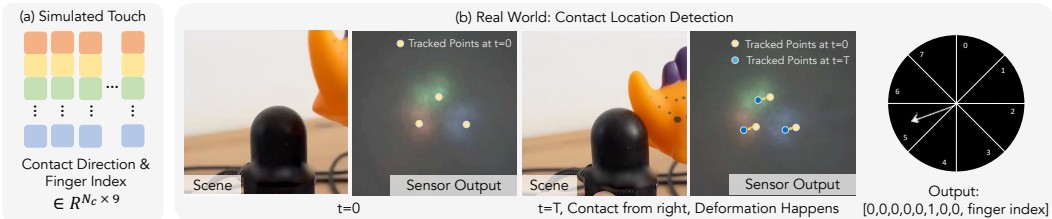

Figure 3: **Representation for Sim-to-Real Touch Sensing**. In the simulation, we use discretized contact location provided by the simulator. In real-world, we detect the deformation by tracking colored regions of the sensor outputs, and parse the same information from a temporal stream of tactile images.

on each sensor, which serves as a proxy for contact position (Figure 3). This 2D keypoint from vision-based touch, similar in spirit to Sodhi et al. [72], is directly fed into the policy.

**Vision (Figure 4).** We use object depth as the vision representation since 1) it is a general representation and does not require human labeling in the real-world and 2) it is hard to realistically simulate RGB images whereas depth is a good abstraction of object shape [35, 73]. In real-world deployment, instead of using the raw depth from a RGBD camera, we use Segment-Anything [12, 13] to segment out the objects to reduce the sim-to-real gap. Formally, given an object depth image $o_t^{\text{depth}}$, we encode it 3-layer ConvNet to output $f_t^{\text{depth}}$. An overview of the vision pipeline is shown in Figure 4. We also randomize the camera position and orientation during training, to make the policy robust to minor viewpoint changes.

**Visuotactile Transformer.** The goal of our visuotactile policy is to accurately infer the learned representation of privileged information. To tackle these challenges, we use a transformer $\phi$ architecture to model these multimodal sensory stream. We concatenate the encoded depth image $f_t^{\text{depth}}$, encoded tactile contact points $f_t^{\text{touch}}$, joint positions $q_t$, and action at the previous timestep $a_{t-1}$ to form the feature vector $f_t$. We feed a sequence of features $f_T = \{f_{t-k}, \ldots, f_{t-1}, f_t\}$ as input

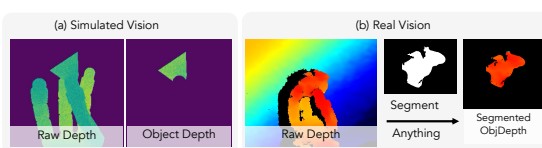

Figure 4: **Representation for Sim-to-Real Vision Sensing** In simulation, we use the object's foreground depth as the input. In real-world, to reduce the sim-to-real gap, we segment out the object's depth map using Segment-Anything.

to the transformer. The transformer outputs $\hat{z}_t$ as the predicted extrinsic vector.

**Training.** Similar to previous work [5, 6, 7], we roll out the *oracle policy* with the *predicted extrinsic vectors* $a_t = \pi(p_t, \hat{z}_t)$ where $\hat{z}_t = \phi(f_T)$. Meanwhile we also store the ground-truth extrinsic vector $z_t$ and construct a training set $\mathcal{B} = \{(f_T^{(i)}, z_t^{(i)}, \hat{z}_t^{(i)})\}_{i=1}^N$. Then we optimize $\phi$ by minimizing the $\ell_2$ distance between $z_t$ and $\hat{z}_t$, and between $a_t$ and $\hat{a}_t$ using Adam [74]. The process is iterated until the loss converges. We apply the same object initialization and dynamics randomization setting.

## 4 Evaluation Setup

**Hardware Setup.** We use an AllegroHand from Wonik Robotics [38] for our experiments. The Allegro hand is a dexterous anthropomorphic robot hand with four fingers, with four degrees of freedom per finger. Position commands are sent to these 16 joints at 20 Hz. The target position commands are converted to torque using a PD Controller at 300 Hz. For depth sensing, we use an Intel RealSense D435 placed at approximately 36cm from the Allegro base. We use an omnidirectional vision-based touch sensor at the distal end of each finger.

**Simulation Setup.** We use the IsaacGym [75] simulator. Each environment contains a simulated AllegroHand and a sampled object from our curated object datasets (Figure 9). Each object is of different physical properties (the exact parameters are in the supplementary material) and a random initial pose. For depth and viewpoint consistency between the real and simulated cameras, we measure the camera-robot extrinsics with an ArUco tag [76] placed on the palm of the real-world Allegro. In

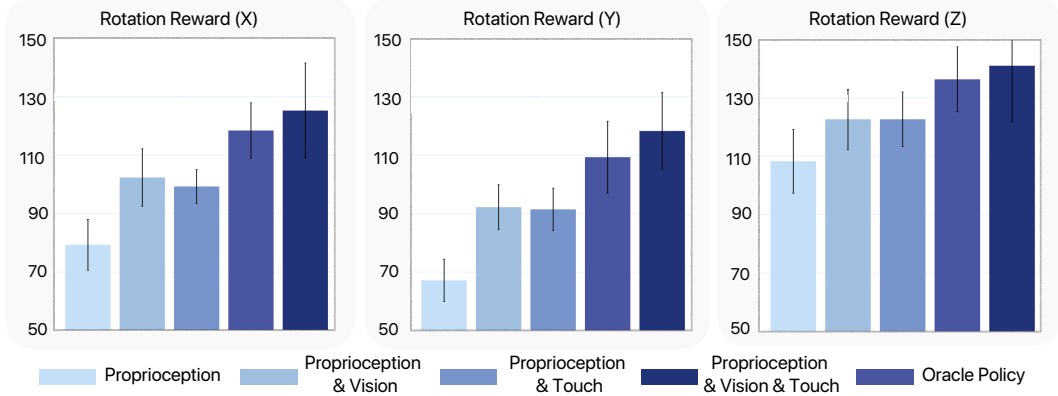

Figure 5: **The importance of vision and touch.** We show the performance improvement of using vision and touch for sensorimotor policy training. Using vision and touch alone can already significantly improve over the proprioception baseline, especially on rotation reward over $x$ and $y$ axis. Combining these two sensings will further improve the performance.

IsaacGym, we use this $SE(3)$ transformation augmented with random pose noise, and further apply realistic depth noise on the resultant images [77].

**Object Set.** We create a curated dataset for objects used in our experiments from EGAD [78], Google Scanned Objects [79], YCB [80], and ContactDB [81]. We select objects with width/depth/height (w/d/h) aspect ratio less than 2.0 (see Figure 9 for a visualization).

**Evaluation Metric.** We use the metrics defined in [7] to evaluate our method both in simulation and in the real-world. In addition, we also evaluate undesired rotation penalties in simulation. We find this metric is particularly important for rotation over $x$ and $y$ axis.

1. *Time-to-Fall (TTF).* The average length of the episode before the object falls out of the hand. This value is normalized by the maximum episode length (20 s).
2. *Rotation Reward (RotR).* This is the average rotation reward $\omega \cdot k$ of an episode in simulation.
3. *Rotation Penalty (RotP).* This is the average rotation penalty per timestep $\omega \times k$.
4. *Radians Rotated (Rotations).* The rotation (in radians) achieved by the policy with respect to the desired axis. This metric is only used in the real world experiments.

## 5 Results and Analysis

In this section, we first quantitatively study our method in simulation. In particular, we study the importance of using object shape information for policy training (Section 5.1), as well as the importance of vision and touch in the visuotactile policies (Section 5.2). Then, we use a shape prediction task to study the information recovered by estimated extrinsic vectors. We show our visuotactile policy learns object shape representation by predicting the 3D shape of objects using $\hat{z}_t$ (Section 5.3). We also evaluate our method on a real-world robot (Section 5.4) and finally show how to train a single policy to rotate over six principle axes.

### 5.1 Object Shape helps Policy Training

The performance is shown in Table 1. We compare *RotateIt* with previous work [7] and our method without the usage of point cloud while still using the quaternion. Experiments show that using point-cloud significantly improves the performance on all of the metrics and for all rotation axis.

To get more insights, we further plot the relative improvements on varies objects shape for $x$-axis rotation, shown in Figure 10 (the "stage1" row). We find that point-cloud gives the largest improvement on objects with non-uniform w/d/h (width/depth/height) ratios and objects with irregular shapes such as the bunny and light bulb. The improvements on regular objects are smaller but still over 40%. In addition, we also evaluate the oracle policy on 15 held-out challenging objects (Figure 6 (b)). We show that not using point cloud results in a 22% decrease in generalization gap while using point-cloud can improve it to only 8% drop. Point-cloud as an input is also used in Qin et al. [82] and Bao et al. [83] but they do not explore how to use it for in-hand manipulation. Note that our

| Touch | x-axis | | | y-axis | | | z-axis | | |
|---|---|---|---|---|---|---|---|---|---|
| | RotR ↑ | TTF ↑ | RotP ↓ | RotR ↑ | TTF ↑ | RotP ↓ | RotR ↑ | TTF ↑ | RotP ↓ |
| Full | $104.29_{\pm10.29}$ | $0.68_{\pm0.04}$ | $0.41_{\pm0.02}$ | $93.05_{\pm9.28}$ | $0.65_{\pm0.01}$ | $0.34_{\pm0.03}$ | $126.73_{\pm10.11}$ | $0.72_{\pm0.03}$ | $0.32_{\pm0.03}$ |
| NoTouch | $79.37_{\pm8.72}$ | $0.46_{\pm0.03}$ | $0.55_{\pm0.02}$ | $67.21_{\pm7.25}$ | $0.48_{\pm0.02}$ | $0.55_{\pm0.03}$ | $108.25_{\pm10.92}$ | $0.62_{\pm0.01}$ | $0.43_{\pm0.02}$ |
| Binary | $80.14_{\pm7.25}$ | $0.47_{\pm0.02}$ | $0.53_{\pm0.03}$ | $66.29_{\pm8.53}$ | $0.49_{\pm0.01}$ | $0.56_{\pm0.04}$ | $110.24_{\pm9.48}$ | $0.63_{\pm0.03}$ | $0.42_{\pm0.02}$ |
| ContactLoc | $102.36_{\pm9.82}$ | $0.65_{\pm0.04}$ | $0.41_{\pm0.04}$ | $92.22_{\pm7.69}$ | $0.64_{\pm0.01}$ | $0.36_{\pm0.03}$ | $122.60_{\pm10.39}$ | $0.73_{\pm0.02}$ | $0.35_{\pm0.01}$ |

Table 2: **The importance of using a finer tactile information.** We compare *RotateIt* which use contact location (ContactLoc) and its variant of using binary contact (Binary) or full contact (position, normal, and scale) information. All methods are without vision information. Binary contact does not provide additional value compared to NoTouch, since it is already contained in our proprioceptive history. We also find using discretized contact locations can match the performance of using full contact in our task.

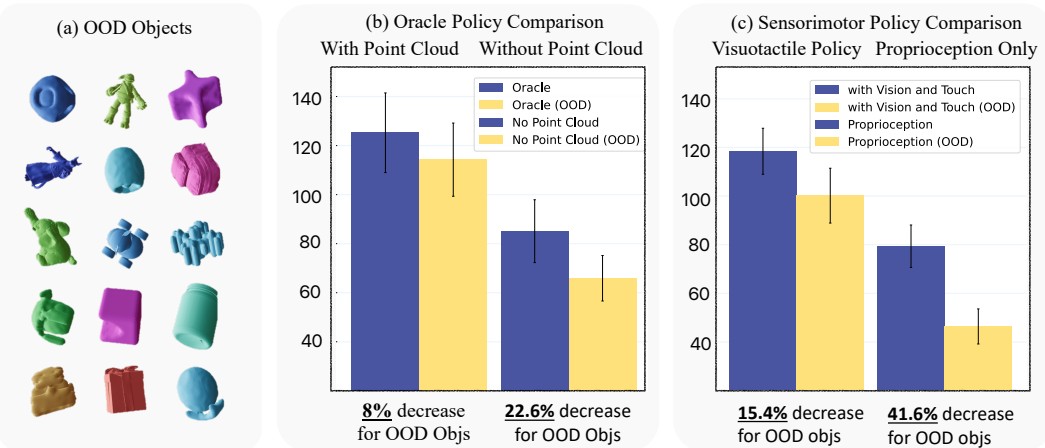

Figure 6: **Out-of-distribution Evaluation.** We evaluate our policies on (a) 15 held-out objects. These objects are also more challenging to manipulate compare to objects in our training set. We show the episode rotation reward for four settings. For oracle policies (b), we find that not using point cloud will lead to 22% performance drop for OOD objects while using point cloud can improve it to only 8%. For sensorimotor policies (c), using proprioception only will lead to 41% drop while visuotactile information can improve it to 15%.

design is different from Chen et al. [8], which uses only pose for the oracle policy and uses object shape information only in the student policy. In our setting using object pose is not sufficient to achieve good enough performance.

## 5.2 Visuotactile Transformer

The oracle policy evaluated in Section 5.1 cannot be transferred to the real-world because it needs access to a manipulated object shape and physical properties. We instead learn to infer this representation during execution from proprioceptive, visual, and tactile history. In Figure 5, we show that using either vision or touch alone gives a significant performance improvements compared to proprioceptive inputs. We also find using a combination of vision and touch sensing can further improve the performance. By integrating visuotactile sensing and temporal transformer, our method can match the performance of the oracle policy. In appendix Table 4, we also show transformer has better sequence modeling ability compared to temporal convolutions used in previous work [5, 7].

In Figure 6 (c), we show the visuotactile information are critical for OOD generalization. Using proprioception only will lead to a 41% performance drop while using vision and touch can improve it to 15% drop.

**Importance of Finer Tactile Sensing.** In contrast to prior work [37], we find in Table 2 that binary contact does not provide benefits. In contrast, contact *locations* are vital for improving performance in *RotateIt*. We speculate that this discrepancy is because Khandate et al. [37] does not use proprioceptive and action history.

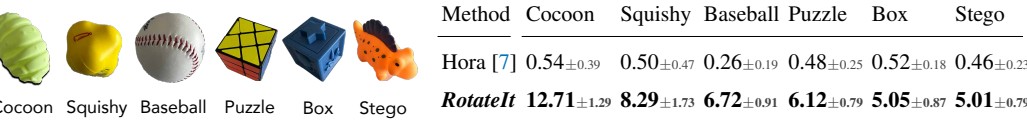

| Method | Cocoon | Squishy | Baseball | Puzzle | Box | Stego |
|---|---|---|---|---|---|---|
| Hora [7] | $0.54_{\pm 0.39}$ | $0.50_{\pm 0.47}$ | $0.26_{\pm 0.19}$ | $0.48_{\pm 0.25}$ | $0.52_{\pm 0.18}$ | $0.46_{\pm 0.23}$ |
| *RotateIt* | $\textbf{12.71}_{\pm 1.29}$ | $\textbf{8.29}_{\pm 1.73}$ | $\textbf{6.72}_{\pm 0.91}$ | $\textbf{6.12}_{\pm 0.79}$ | $\textbf{5.05}_{\pm 0.87}$ | $\textbf{5.01}_{\pm 0.79}$ |

Figure 8: **Rotations rotated (↑) over $x$-axis for *RotateIt* and [7] in Real-world Evaluation.** We compare *RotateIt* and Hora [7] on six different objects. Hora [7] is not able to finish this task and does not learn finger-gaiting to rotate the object, while *RotateIt* can.

## 5.3 Representation Learned in the Latent Space

Next, we study the information that is encoded into $z_t$ and $\hat{z}_t$. After we finish training the four policies in Figure 7, we freeze the network and then we run our policy on 20 objects in our object dataset (16 for training, 4 for testing). This gives us an extrinsic vector dataset for each policy. On each of the datasets, we train one decoder whose input is a sub-sequence of extrinsic vectors and output is the voxel grid. After training this decoder, we run it on the 4 held-out testing objects.

In Figure 7, we visualize predicted shapes averaged over 100 randomly selected subsequences from roll-outs on novel test objects for four policies: the stage 1 oracle policy with and without shape (mesh) conditioning, and the stage 2 policy with and without visuotactile sensory inputs. The results suggest that

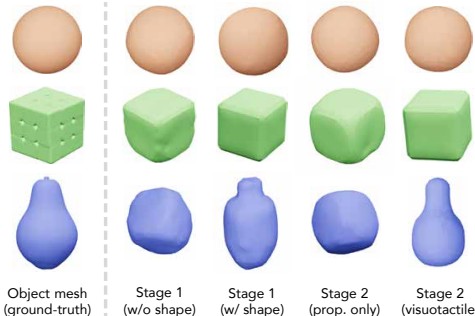

Figure 7: **Inverting encoded extrinsics.** We predict 3D shapes on novel objects from learned $z_t$ and $\hat{z}_t$. Stage 1 results are provided with / without shape conditioning, stage 2 results are provided with / without visual and tactile sensors.

shape information is preserved and useful for our oracle policy even though the only learning signal is the reward function. Next, our results also highlight both the capabilities and limits of proprioception, which we see can robustly distinguish between spherical (beige) and cuboidal (green) objects. Shape understanding for more irregular objects like the pear (blue), however, requires additional sensors. This supports the increased benefit of vision and touch for more complex objects that we observe in Section 5.1.

## 5.4 Real-world Evaluations

Finally, we quantitatively compare *RotateIt* and *Hora* [7] in the real-world on rotating different objects over the $x$-axis. We find that without vision and touch, *Hora* cannot finish this task. It only learns in-grasp movement with thumb slowly moving to the bottom of the object. It is also not able to maintain stability; the object quickly falls down. In contrast, *RotateIt* can successfully manipulate multiple objects with different geometries such as cubes, spheres, or cylinders by ∼$2\pi$ radians within 20 seconds. Note that many real-world objects are outside our training set such as the box, Cocoon, Squishy, and Stego. The real-world physics is also different from the simulated physics. Having a successful sim-to-real transfer is a strong evidence of generalization. We show qualitative results on rotation around and beyond the three canonical axes on our website. In the video, we also test a policy trained with the assumption that one of the touch sensors is off. The policy performs similarly to the full policy, demonstrating the robustness of the algorithm.

## 6 Limitations and Future Work

In this paper, we show the feasibility of training policies that can rotate many objects over multiple axes. We view this capability as an important step towards general-purpose in-hand manipulation. We assume the objects are not too long (e.g. a pencil or a screwdriver) and are within the mechanical limit of the robot hand. Our method is not able to utilize real-world experiences during deployment since it is frozen after training. There are also various ways to improve the touch processing system since we only use the low-dimensional contact location as the input and do not utilize the full information output by the omnidirectional image-based tactile sensor.

**Acknowledgments**

This research was supported as a BAIR Open Research Common Project with Meta. In their academic roles at UC Berkeley, Haozhi Qi and Jitendra Malik are supported in part by DARPA Machine Common Sense (MCS), Brent Yi is supported by the NSF Graduate Research Fellowship Program under Grant DGE 2146752, and Haozhi Qi, Brent Yi, and Yi Ma are partially supported by ONR N00014-22-1-2102 and the InnoHK HKCRC grant. Roberto Calandra is funded by the German Research Foundation (DFG, Deutsche Forschungsgemeinschaft) as part of Germany's Excellence Strategy – EXC 2050/1 – Project ID 390696704 – Cluster of Excellence "Centre for Tactile Internet with Human-in-the-Loop" (CeTI) of Technische Universität Dresden. We thank Shubham Goel, Eric Wallace, and Angjoo Kanazawa, Raunaq Bhirangi for their feedback. We thank Austin Wang and Tingfan Wu for their help on hardware. We thank Xinru Yang for her help on real-world videos.

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

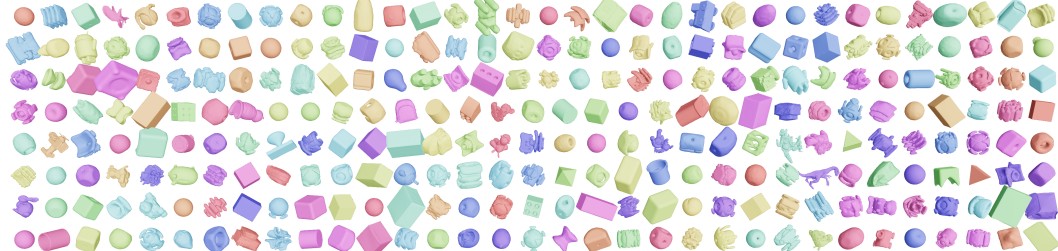

Figure 9: **Training objects**. We curated a diverse combination of objects from EGAD [78], Google Scanned Objects [79], YCB [80], and ContactDB [81]. We filter out meshes with disconnected components and objects with a width/depth/height (w/d/h) ratio larger than 2.0.

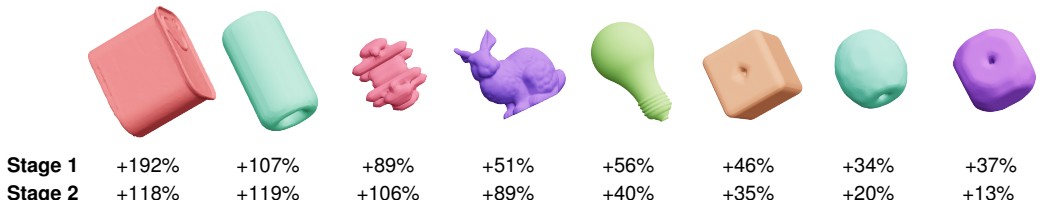

| | | | | | | | | |
|---|---|---|---|---|---|---|---|---|
| **Stage 1** | +192% | +107% | +89% | +51% | +56% | +46% | +34% | +37% |
| **Stage 2** | +118% | +119% | +106% | +89% | +40% | +35% | +20% | +13% |

Figure 10: **Relative rotation reward improvements before and after shape or visuotactile information.** For stage 1 training (oracle policy), we compare our oracle policy and the policy without point-cloud as input. For stage 2 training (visuotactile policy), we compare the improvement of *RotateIt* and the policy with only proprioceptive input. In both cases, having vision and touch information significantly improve the performance.

## A  Additional Experiments

### A.1  Relative Improvements on Different Objects

We also find policies without object shape will consider all the objects as spherical or cuboid objects, which explains the huge improvement on objects with large w/d/h ratios Figure 10.

Similar to what we find in the oracle policy training, we observe the visuotactile policy has larger improvements on irregular and non-uniform objects (Figure 10, "stage2" row).

### A.2  Multi-axis Training

In previous sections, each oracle policy is trained with a fixed rotation axis $k$. In this section, we demonstrate it is also feasible to train a single network to perform multi-axis object rotation. To achieve this, we augment the observation space with $k$ and train it with the reward defined in Section 3.1 and the imitation learning objective with the corresponding single-axis oracles.

| Method | $+x$ | $-x$ | $+y$ | $-y$ | $+z$ | $-z$ |
|---|---|---|---|---|---|---|
| Single-axis | $110.19_{\pm 8.26}$ | $104.29_{\pm 10.29}$ | $93.05_{\pm 9.28}$ | $90.20_{\pm 10.39}$ | $124.91_{\pm 8.78}$ | $126.73_{\pm 10.11}$ |
| Multi-axis | $105.21_{\pm 9.27}$ | $103.11_{\pm 10.17}$ | $85.38_{\pm 9.71}$ | $89.83_{\pm 10.11}$ | $125.32_{\pm 7.81}$ | $125.19_{\pm 9.93}$ |

Table 3: Episode Rotation Reward comparison between single-axis training and multi-axis training. The distilled multi-axis policy performs on par with the single task oracles.

We show the episode rotation reward for both the single-axis oracle policy and the multi-axis policy in Table 3. We empirically find the distilled multi-axis policy performs on par with the single task oracles. We also observe the policy does not converge when training with only reinforcement learning.

### A.3  Detailed Comparison of Using Vision and Touch.

Table 4 shows the detailed comparison of using vision, touch, and the transformer architecture. We show each of the component can significantly improve over the baseline and are also complement with each other.

### A.4  Randomization of Simulated Vision Sensing.

During training, we apply various randomizations to the vision sensing to make it robust. We evaluate our model under different noise setting in simulation. The results are shown in Table 5.

| | Modality | | x-axis | | | y-axis | | | z-axis | | |
|---|---|---|---|---|---|---|---|---|---|---|---|
| Method | Vision | Touch | RotR ↑ | TTF ↑ | RotP ↓ | RotR ↑ | TTF ↑ | RotP ↓ | RotR ↑ | TTF ↑ | RotP ↓ |
| Oracle | N/A | N/A | $125.23_{\pm16.24}$ | $0.79_{\pm0.03}$ | $0.35_{\pm0.02}$ | $118.26_{\pm13.20}$ | $0.79_{\pm0.05}$ | $0.30_{\pm0.01}$ | $140.90_{\pm19.26}$ | $0.82_{\pm0.02}$ | $0.27_{\pm0.01}$ |
| Conv | | | $66.23_{\pm8.72}$ | $0.41_{\pm0.04}$ | $0.64_{\pm0.01}$ | $54.19_{\pm9.27}$ | $0.38_{\pm0.02}$ | $0.69_{\pm0.02}$ | $89.21_{\pm12.37}$ | $0.56_{\pm0.03}$ | $0.47_{\pm0.03}$ |
| | ✓ | | $87.21_{\pm12.11}$ | $0.59_{\pm0.02}$ | $0.59_{\pm0.02}$ | $72.51_{\pm9.10}$ | $0.57_{\pm0.03}$ | $0.62_{\pm0.03}$ | $102.35_{\pm10.74}$ | $0.68_{\pm0.02}$ | $0.42_{\pm0.01}$ |
| | | ✓ | $82.19_{\pm9.21}$ | $0.60_{\pm0.03}$ | $0.57_{\pm0.01}$ | $69.99_{\pm7.26}$ | $0.58_{\pm0.01}$ | $0.61_{\pm0.02}$ | $107.73_{\pm9.83}$ | $0.63_{\pm0.02}$ | $0.46_{\pm0.02}$ |
| | ✓ | ✓ | $98.20_{\pm10.18}$ | $0.70_{\pm0.03}$ | $0.45_{\pm0.03}$ | $89.82_{\pm9.22}$ | $0.67_{\pm0.03}$ | $0.47_{\pm0.01}$ | $113.26_{\pm13.98}$ | $0.70_{\pm0.04}$ | $0.40_{\pm0.01}$ |
| Transformer | | | $79.37_{\pm8.72}$ | $0.46_{\pm0.03}$ | $0.55_{\pm0.02}$ | $67.21_{\pm7.25}$ | $0.48_{\pm0.02}$ | $0.55_{\pm0.03}$ | $108.25_{\pm10.92}$ | $0.62_{\pm0.01}$ | $0.43_{\pm0.02}$ |
| | ✓ | | $102.36_{\pm9.82}$ | $0.65_{\pm0.04}$ | $0.41_{\pm0.04}$ | $92.22_{\pm7.69}$ | $0.64_{\pm0.01}$ | $0.36_{\pm0.03}$ | $122.60_{\pm10.39}$ | $0.73_{\pm0.02}$ | $0.35_{\pm0.01}$ |
| | | ✓ | $99.29_{\pm5.79}$ | $0.62_{\pm0.05}$ | $0.43_{\pm0.03}$ | $91.47_{\pm7.26}$ | $0.60_{\pm0.02}$ | $0.37_{\pm0.02}$ | $125.24_{\pm9.32}$ | $0.72_{\pm0.03}$ | $0.39_{\pm0.04}$ |
| | ✓ | ✓ | $\mathbf{118.42_{\pm9.46}}$ | $\mathbf{0.75_{\pm0.03}}$ | $\mathbf{0.37_{\pm0.02}}$ | $\mathbf{109.31_{\pm12.29}}$ | $\mathbf{0.73_{\pm0.02}}$ | $\mathbf{0.31_{\pm0.04}}$ | $\mathbf{136.25_{\pm11.12}}$ | $\mathbf{0.80_{\pm0.04}}$ | $\mathbf{0.29_{\pm0.02}}$ |

Table 4: **The importance of vision and touch.** We show the performance improvement of using vision, touch, and the transformer architecture. Each of the three components significantly improves the performance of rotating over $x/y/z$ axis.

We add gaussian noise to camera positions and orientations. Cam Pos stands for the value for each different setting (in meters). Cam RPY stands for the extend we randomize the camera rotation (in roll/pitch/yaw values, in radius). The camera field-of-view (fov) is also randomized. The values are set to a uniform distribution according to the Cam FOV column. We also simulate segmentation noise (for each pixel, with probability $p$, the mask is flipped) and segmentation failure (for each timestep, with probability $p$, the mask is completely 0) to simulate segmentation errors in the real-world.

We find that the model behaves robustly under training randomization and slightly out-of-distribution noises. However, too large noise will still impact the performance, highlighting the importance of proper camera calibration.

## B Implementation Details

**Simulation Setup** During training, we use 32768 parallel environments to collection samples for training the agent, distributed on 4 GPUs. Each environment contains a simulated AllegroHand and a sampled objects from our curated object datasets (Figure 9). Each object is of different physical properties and a random initial pose. The simulation frequency is 200 Hz and the control frequency is 20 Hz. Each episode lasts for 400 control steps (equivalent to 20 s). We reset the episode if the objects fall below 13.5 cm with respect to the hand.

**Stable Precision Grasp Generation.** Our approach assumes the object is grasped at the beginning of the episode. To achieve this, we start from a canonical grasping position using fingers. Then we add a relative offset to the joint positions sampled from $\mathcal{U}(-0.25, 0.25)$rad. Then we forward the simulation by 0.5 s. We save the grasping pose if all the following conditions are satisfied:

1. The distance between the fingertip and the object should be smaller than 10 cm.
2. At lease two fingers are in contact with the object.
3. The object's height should be above 13.5 cm higher than the center of the palm.

In practice, we discretized (each region is separated by 0.2) the scales specified in Table 6 and pre-sampled 400 grasping poses for each object and for each scale.

**Physical Randomization Parameter.** We apply domain randomization during training the oracle policy as well as the visuotactile policy. The parameters are listed in Table 6. Following [1], we apply a random disturbance force to the object during training whose scale is $2m$ where $m$ is the object mass. The force is decayed by 0.9 every 80 ms following [1]. The force is re-sampled at each time-step with a probability 0.25.

**Reward Hyperparameter.** We use $r_{\max} = 0.5$, $r_{\min} = -0.5$, $\lambda_{\text{torque}} = -0.1$, $\lambda_{\text{linvel}} = -0.3$, $\lambda_{\text{work}} = -2.0$, and $\lambda_{\text{rotp}} = -0.1$. We also find that if we apply $\lambda_{\text{rotp}} = -0.1$ at the start of training, the policy will only learn to stably hold the objects. Therefore we set this coefficient to be 0 at the beginning and then linearly decrease it to $-0.1$ using curriculum learning [84].

| Setting | Cam Pos | Cam RPY | Cam FOV | Seg Noise | Seg Failure | RotR ↑ |
|---|---|---|---|---|---|---|
| Perfect Vision | 0 | 0 | 0 | 0 | 0 | 119.19 |
| Same Noise as training | $+\mathcal{N}(0, 0.01)$ | $+\mathcal{N}(0, 0.03)$ | $=\mathcal{U}(52, 58)$ | 0.2 | 0.05 | 118.42 |
| Out-of-distribution Noise | $+\mathcal{N}(0, 0.015)$ | $+\mathcal{N}(0, 0.035)$ | $=\mathcal{U}(48, 62)$ | 0.25 | 0.075 | 115.30 |
| Larger Noise | $+\mathcal{N}(0, 0.02)$ | $+\mathcal{N}(0, 0.04)$ | $=\mathcal{U}(45, 65)$ | 0.3 | 0.1 | 102.80 |
| No Vision | / | / | / | / | / | 99.29 |

Table 5: **Evaluation performance on noisy vision sensing systems.** We evaluate the same visuotactile policy on five different settings. We find that the model behaves robustly under training randomization and slightly out-of-distribution noises. However, too large noise will still impact the performance, highlighting the importance of proper camera calibration.

| Parameter | Range |
|---|---|
| Object Scale | [0.46, 0.68] |
| Mass | [0.01, 0.25] kg |
| Center of Mass | [-1.00, 1.00] cm |
| Coefficient of Friction | [0.3, 3.0] |
| External Disturbance | (2, 0.25) |
| PD Controller Stiffness | [2.9, 3.1] |
| PD Controller Damping | [0.09, 0.11] |

Table 6: Randomization Range of Physics Parameters. We sample the physical parameter values from a uniform distribution.

| Parameter | Default Value |
|---|---|
| $N_p$ | 100 |
| $c_p$ | 32 |
| $\dim(z_t^{shape})$ | 32 |
| $\dim(z_t^{phys})$ | 8 |
| $\dim(z_t)$ | 40 |
| $\dim(f_t^{depth})$ | 32 |
| $\dim(f_t^{touch})$ | 32 |

Table 7: **Default Values for network hyperparameters**.

**Network Architecture.** The oracle control policy $\pi$ is a multi-layer perceptron (MLP) which takes in the state $p_t \in \mathbb{R}^{96}$ and the embedding of privileged information $z_t \in \mathbb{R}^{40}$, and outputs a 16-dimensional action vector $a_t$. There are four layers with hidden unit dimension [512, 256, 128, 16]. We use ELU [85] as the activation function. The privileged encoder $\mu$ is also a three layer MLP with hidden unit dimension [256, 128, 8] and encodes object pose and physical properties to output $z_t^{\text{phys}} \in \mathbb{R}^8$. We use ReLU as the activation function. The PointNet Encoder is a three layer MLP with hidden unit dimension [32, 32, 32]. The MLP is applied on each of points and then the features are aggregated using max pooling.

The visuotactile transformer takes object depth feature, touch feature, proprioception feature, and action history as input. The object depth image is of size $60 \times 60$ and is first be passed to a four layer ConvNet and then a global average pooling layer to produce the feature of dimension 32. The contact location is a 9-dimension vector and is first passed to an MLP with hidden unit dimension [32, 32, 32]. The contact feature is aggregated using average pooling. For the robot joint position and actions, we first encode them into a 32-dimensional representations for each timestep via a two-layer MLP (with hidden unit dimension [32, 32]). The feature dimension of our transformer is 32 and with depth 2. The self-attention module has 2 parallel head.

**Optimization Details.** During the oracle policy training, we jointly optimize the control policy $\pi$ and the privileged information encoder using PPO [71]. In each PPO iteration, we collect samples from 32768 environments with 10 agent steps each (corresponding to 0.5 seconds). We train 5 epochs with a batch size 32,768. The learning rate is $5e-3$. For the visuotactile policy, we use Adam optimizer [74] to minimize MSE loss. The learning rate is $3e-4$.

