# OpenReview forum: "General In-hand Object Rotation with Vision and Touch"
_robot-learning.org/CoRL/2023/Conference — CoRL 2023 Poster_

### Official Review · Reviewer_3t8K · 2023-07-16

**Confidence:** 5
**Originality:** Good
**Technical Quality:** Good
**Clarity Of Presentation:** Good
**Impact:** 2

**Recommendation:**

Weak Reject: I recommend rejecting the paper, but will not argue for my recommendation if the majority of other reviewers have a different opinion.

**Review:**

I summarize the strengths and weaknesses below.

**Strengths**:

1. The paper is clear and the technical part is easy to follow.

2. The experiments effectively demonstrate the advantages of the proposed approach over the compared baselines.

3. The real-world demo in the video shows convincing results on the sim-to-real performance of the approach.

**Weaknesses**:

1. It lacks motivation for the paper and the approach. Given the existing works (especially Chen et. al. [8]), it is unclear to me what aspects this paper tries to solve beyond the existing work's scope and why the proposed technique (e.g. RMA framework, touch information) helps address the goal.

2. Continued from the last question, it is also unclear to me about the advantage of the proposed approach over the existing work (Chen et al. [8])? The single-axis rotation task seems to be a simpler task than the random goal pose re-orientation task solved by [8]. As a suggestion, it would make this paper much more valuable if it can demonstrate some advantages over [8] such as either having higher performance or being more learning efficient on random object random goal pose re-orientation tasks.

3. It seems that the trained policy only works for a fixed axis instead of goal conditioning on a random axis.

4. It is unclear whether the trained policy is able to generalize to unseen objects.

**Quality Of The Limitations Section:**

Additional details required

**Questions For Rebuttal:**

I have the following questions and suggestions that I hope the authors can clarify and address during the rebuttal:

1. Was the proposed method tested on unseen objects? The input of object mesh might help perform better on trained objects, but it might also result in overfitting to seen objects.

2. Does the trained policy work for an arbitrary axis or each policy is trained just for one single axis? It is unclear to me in the paper.

3. Line 79 mentions that one advantage of the proposed method is that it can manipulate texture-less and symmetrical objects. Is this indeed an advantage of the simpler task of rotating around an axis so that there is no need to determine the success/termination of the task based on the object pose? For instance, can we apply the proposed approach to reorient texture-less objects?

4. The proposed approach is mainly based on RMA approach. It would be more appropriate if the author could give more credit to RMA paper in the method section.

5. For the experiment section 5.3, the authors show that it is able to recover the full geometry from the latent $z$. Did the authors train a decoder separately? It lacks details about how this reconstruction work in the paper. Also it seems that there is no reconstruction loss during training, so what is the secret sauce that encourages the latent space to contain meaningful info to reconstruct the mesh?

6. Line 151: is the 9-dim vector for a single finger or for the full hand?

**Robotics Focus:**

Sufficient demonstration on hardware

**Summary Of Paper:**

This paper presents a policy learning method for in-hand object rotation. The goal is to learn a general and realistic policy that can rotate any object around a fixed axis as fast as possible in simulation and then transfer to real world. The method is based on RMA approach. It first learns an expert policy that acts with input privilege state information, object physical parameters, and object point cloud. Following RMA, the learned expert policy is composed of two parts, a task-relevant encoder, and a control-relevant policy layer. It then distills the expert policy to a policy with real observation as input by learning a new task-relevant encoder to map the history of observations into the corresponding latent space. It extends the previous RMA-based approach [7] by using visual (i.e. depth image) and touch (i.e. contact location) to infer the latent task-related representation. The conducted experiments show the effectiveness of the approach.

**Summary Of Recommendation:**

While this paper shows a great system for object rotating and demonstrates the results in both simulation and real world, given the listed questions and concerns, I cannot fully assess the technical significance of this paper over the existing works. Therefore I am towards the negative side of the paper for now.

---

### Official Review · Reviewer_ahST · 2023-07-18

**Confidence:** 4
**Originality:** Good
**Technical Quality:** Very Good
**Clarity Of Presentation:** Very Good
**Impact:** 3

**Recommendation:**

Weak Accept: I recommend accepting the paper, but will not argue for my recommendation if the majority of other reviewers have a different opinion.

**Review:**

The quality, originality, and significance of the paper are good. The contributions and technic details are clearly presented in the paper. The literature is well reviewed and organized to understand the contributions of the work. The framework and training pipeline are defined clearly both in diagram and math notation. Results are also reported and compared with relevant existing methods, which help the reader to understand the difference easily. The weaknesses are listed in the issues.

**Quality Of The Limitations Section:**

Limitations are addressed clearly

**Questions For Rebuttal:**

Issues:
1.	It seems the system encodes quite abundant of information into the extrinsics vector including shape from point cloud and physical property. While this makes the system working well in the rotating tasks, it makes the system more complex. I’m wondering if this helps or limit the generalization ability of the system to novel objects. Although the authors show the representation learned of 3 novel objects, these shapes are very common and not very different from the shapes in the training data. How does the system performs on rotating objects with distinct shapes?
2.	How does the accuracy of the vision sensing from the depth camera affect the performance of the system? The depth image in Figure 4(b) looks very rough and has occlusion from the robotic hand.


**Robotics Focus:**

Sufficient demonstration on hardware

**Summary Of Paper:**

The paper proposes a system capable of general fingertip-based in-hand object rotation with a dexterous hand. The system uses both visual and tactile sensing as input, which significantly improves the manipulation performance. Meanwhile, the latent representation of the policy captures the 3D shapes of objects and works as primitives for general dexterous manipulation.

**Summary Of Recommendation:**

The paper presents a system capable of general in-hand object rotation using vision and tactile sensing. The idea is interesting to the community and the presentation is well organized. Hence, I recommend accepting the paper.

---

### Official Review · Reviewer_Kkn8 · 2023-07-19

**Confidence:** 3
**Originality:** Good
**Technical Quality:** Good
**Clarity Of Presentation:** Good
**Impact:** 3

**Recommendation:**

Weak Accept: I recommend accepting the paper, but will not argue for my recommendation if the majority of other reviewers have a different opinion.

**Review:**

The paper indeed presents impressive results, particularly regarding multi-axis in-hand reorientation, sim2real outcomes, and the latent space's capability to recover 3D object geometry. Aligning the latent space (z_t) learned from simulation, which has access to all privileged information, with the real-world latent space (\hat{z}_t) obtained from limited information is a commendable approach. Additionally, the ablations and comparisons with their baseline (Hora) add to the paper's strengths.

However, my main concern about this work is their real-world result, where one of the four touch sensors was broken at the time of submission. This is problematic because their result will not match the simulation from training and what they described in the methodology (line 154). I appreciate the author's honesty about this issue, but I think they should have withheld their submission until they fixed their hardware and obtained a non-misleading real-world result.

Furthermore, there are some mistakes in the manuscript and figures. I left details in the following section.

**Quality Of The Limitations Section:**

Limitations are addressed clearly

**Questions For Rebuttal:**

* Is it possible to fix your hardware such that it matches your simulate environment and your methodology descriptions?
* To improve the manuscript, the author could consider the following:
1. Enhance clarity by explicitly mentioning the dimensions of each input in section 3.1.
2. In Figure 4 (c), what does '!' mean?
3. Figure 5 "we our oracle policy and our policy"

**Robotics Focus:**

Sufficient demonstration on hardware

**Summary Of Paper:**

This paper focuses on in-hand object rotation using fingertips, utilizing both vision and tactile information. The main contributions of this paper are as follows:

1. Emphasizing the significance of combining both vision and tactile inputs for effectively performing the in-hand re-orientation task.
2. Proposing a novel latent manipulation policy that leverages a latent space to implicitly capture the geometrical understanding of objects without requiring direct supervision.
3. Conducting direct sim2real experiments on multiple different objects and axes to validate the proposed approach's effectiveness.

**Summary Of Recommendation:**

This work includes Interesting result regarding in-hand re-orientation with vision and touch. I am willing to increase the recommendation if the author update the result with fully working hardware during the rebuttal.

---

### Official Review · Reviewer_Ms7j · 2023-07-21

**Confidence:** 3
**Originality:** Good
**Technical Quality:** Very Good
**Clarity Of Presentation:** Excellent
**Impact:** 4

**Recommendation:**

Strong Accept: I recommend accepting the paper and will argue for my recommendation even if other reviewers hold a different opinion.

**Review:**

<Strengths>

1. Overall, this is a strong empirical systems paper that does many scientific investigations and ablation studies on the effect of various decisions to doing in-hand manipulation within the teacher-student framework.

2. The hardware experiments are extensive and convincing, and the authors do a good job of implementing the system.

<Weaknesses>

1. The parts of the hands that are used to make contact are quite minimal, although this might likely be due to the difficulty of installing tactile sensors among the surface of the hand.

2. Although the technical and scientific contribution of the paper is quite strong, the fundamental methods used in the paper follow the same algorithm, so the algorithmic contribution / novelty is quite low.


**Quality Of The Limitations Section:**

Limitations are addressed clearly

**Questions For Rebuttal:**

1. Would a similar strategy work for in-hand translation / commanding general pose commands that combine translation and rotation?

2. Do the author’s proposed method works for object outside the trained dataset? What is the extent to which it generalizes across shapes?


**Robotics Focus:**

Sufficient demonstration on hardware

**Summary Of Paper:**

The authors tackle object rotation with a teacher-student framework, where PPO is trained with privileged information in the simulator first, and the student is forced to match the encoded extrinsics given observation data. The authors consider certain design choices that are unique compared to previous work: depth is used for training instead of RGB, and dense tactile signals on locations are used. The authors show that these series of choices lead to much improved performance compared to previous works that do vision-only or touch-only.

**Summary Of Recommendation:**

The paper scientifically investigates many of the design choices behind the teacher-student PPO framework within in-hand manipulation, and the results from this investigation can be very informative for other researchers doing in-hand manipulation. The presentation is thorough and the paper is written well, and all the investigations are scientifically done by comparing against appropriate baselines, so I recommend acceptance.

---

### Author Response · Authors · 2023-08-15
**Shared Response to all reviewers**

**Q1: Generalization outside the trained dataset (Ms7j, ahST, 3t8K).**

A1: We want to clarify that many real-world objects are outside our training set such as the cube with holes or the dinosaur toy. The real-world physics is also different from the simulated physics. Having a successful sim2real transfer is a strong evidence of generalization. We also updated the manuscript to emphasize this fact.

In addition, to demonstrate shape information (in oracle policy) and visuotactile information (in visuotactile policy) are helpful for generalization, we conduct simulation experiments on [15 novel held-out objects](https://sites.google.com/view/rotateit).
We show the average episode rotation reward for four different settings both on our training set and novel objects set. The novel object set contains out-of-distribution and objects with more challenging shapes compared to what is used in training. The results are as follows:

| Method                                   | Within Training Distribution         | Novel and Challenging Objects       | Relative Decrease     |
|-------------------------------------------------|--------------|-----------------|-----------------|
| Oracle, without point-cloud | 85.10 | 65.90 | 22.6% |
| **Oracle, with point-cloud** | 125.23 | 114.26 | **8.0%** |
| Proprioception | 79.37 | 46.38 | 41.6% |
| **Visuotactile** | 118.42 | 100.17 | **15.4%** |

For oracle policy, not using point cloud results in a 22% decrease in generalization while using point-cloud can improve it to only 8% drop.

A similar trend can also be observed in visuotactile policy. Using proprioception only will lead to a 41% performance drop while using vision and touch can improve it to 15% drop. This is quite small, and indicative of good OOD performance

---

### Author Response · Authors · 2023-08-15
**Message to AC**


**Edit**: We would like update the AC that the reviewer 3t8K has now [positively responded](https://openreview.net/forum?id=RN00jfIV-X&noteId=jsIdFAFxqf-) to our clarification, but we will keep the below comments for completeness.

---

Dear AC,

We want to raise our concerns about the reviews from reviewer 3t8K.

---

**1. The main critique from reviewer 3t8K is our difference from Chen et al [8], which is very clear in our opinion.**

The reviewer 3t8K requests to apply our method to do in-hand reorientation, which is a very different task from continuous rotation. Continuous rotation ideally goes for many revolutions (multiples of 2\pi). Asking us to solve the reorientation problem should be regarded as out-of-scope.

The other requests such as training a multi-task policy and rotation policy over non-canonical axes have been addressed in our rebuttal.

Compared to Chen et al. the sensing input to the visuotactile policy and the input to oracle policies are all quite different and important to the task. We have done comprehensive investigations on the effects of using point-cloud and touch sensing. This is also acknowledged by other reviewers such as Ms7j "strong empirical systems paper that does many scientific investigations and ablation studies on the effect of various decisions" and "hardware experiments are extensive and convincing".

The differences are summarized below:

|                                   | Chen et al. [8]         | Ours       |
|-------------------------------------------------|--------------|-----------------|
| Task                         | Reorientation | Continuous Rotation |
| Sensing                          | No Tactile Sensing | Tactile Sensing |
| Hardware | Customized Symmetric Fingers | AllegroHand|
| Oracle Vision | Quaternion | Object Shape (Point-Cloud) |

---

**2. Note that Chen et al. [8] has not yet been published.**

Best,

Paper 304 Authors

---

### Decision · Program_Chairs · 2023-08-30

**Decision:**

Accept (Poster)

**Comment:**

The paper presents learning-based control method for in-hand manipulation using depth and tactile information. The method is somewhat common (i.e., teacher-student pipeline) but it is the first application to this problem.

Many reviwers agreed that the method is not new. The technical contribution including the experiments are strong but the weakside is on the novelty of the method.

I accept the fact that the comparisons to Chen et al.[8] is not important now because the paper is not published. The use of tactile information also differentiates the two papers. As I mentioned, the algorithmic contribution is limited in this paper so algorithmic comparisons are not important in my opinion.

The new experiments on held-out objects make this paper stronger. But these have to be added to at least in the supplementary materials.

I belive that the weakness (weak novelty, rotate only in canonical axes) and strength (comprehensive experiments, good integration) are clear. Since all reviwers have positive opinions, I am also willing to accept this paper.